# Biomolecules as Flame Retardant Additives for Polymers: A Review

**DOI:** 10.3390/polym12040849

**Published:** 2020-04-07

**Authors:** Daniel A. Villamil Watson, David A. Schiraldi

**Affiliations:** Department of Macromolecular Science and Engineering, Case Western Reserve University, Cleveland, OH 44106, USA; dav31@cwru.edu

**Keywords:** bio-based, flame retardants, thermal stability, protein, tannins

## Abstract

Biological molecules can be obtained from natural sources or from commercial waste streams and can serve as effective feedstocks for a wide range of polymer products. From foams to epoxies and composites to bulk plastics, biomolecules show processability, thermal stability, and mechanical adaptations to fulfill current material requirements. This paper summarizes the known bio-sourced (or bio-derived), environmentally safe, thermo-oxidative, and flame retardant (BEST-FR) additives from animal tissues, plant fibers, food waste, and other natural resources. The flammability, flame retardance, and—where available—effects on polymer matrix’s mechanical properties of these materials will be presented. Their method of incorporation into the matrix, and the matrices for which the BEST-FR should be applicable will also be made known if reported. Lastly, a review on terminology and testing methodology is provided with comments on future developments in the field.

## 1. Introduction

### 1.1. History of Fire Retardants

While intumescence is a leading scientific endeavor in fire protection engineering in modern times, its discovery dates back to British Patent 551, granted in 1735 to Obadiah Wyld [1], which set the precedent for the boom seen in the 1970s–1980s flame retardant (FR) industry—which consequently lead to the current fast-replacement-based industry of fire protection additives. FRs can be qualified as either: additive, those assimilated into the polymer matrix during processing; or reactive, additives directly fixated to the polymer chain during synthesis or as a secondary (post) reaction. Several hallmark inventions preceded this robust industry. Gay Lussac in 1821 could be considered as a “grandfather” figure for conceptual bio-based flame resistant fabrics with a hemp-flax weave treated with borate and APP (ammonium polyphosphate) [2]. In 1934, a German patent for DAP (diammonium phosphate) and formaldehyde wood treatment marked the first patented intumescent FR—albeit the written claim to “intumescent” belongs to the 1938 patent of Tramm et al. [3]. Slow and scattered development continued with fireworks and various mixtures, such as nitrated linseed oil and naphthalene. Such was the disarray of available information, and not until the middle of WWII did intumescents become a priority for the US Navy [4]. Thus, the definition of an intumescent system emerged prescribing the following components: an acid source and/or catalyst, a carbonic, a spumific, and a binder [5]. APP quickly gained prominence; soon APP-II with a higher degree of polymerization (chains length >1000 *ν*), *T*_d_ of 280 °C, inert ammonia gaseous product, and polyphosphoric acids glassy surface residue became ubiquitous in the field of FR [5]. Over time it has become apparent that intumescents are critical in protecting structural steel from the 550 °C critical mechanical loss limit [6].

Throughout the past 40 years, the most prevalent and efficient FRs have been identified as HFR (halogenated flame retardants), P, N, Al, Mg, B, At, Mb, and some nanofillers [7]. Several testing methodologies and technologies have also emerged to qualitatively and (to a lesser extent) quantitatively, measure and compare FR efficacies. Leading experimental techniques include: limiting-oxygen index (LOI; ISO 4589); UL–94 arrangements; cone calorimetry (CC); and thermo-gravimetric analysis (TGA) coupled with, but not limited to, FT-IR (Fourier transform infrared spectroscopy), EGA (evolved gas analysis), MS (mass spectroscopy), and GC (gas chromatography) [7,8]. From fire-fighters to police officers, the treatments which have seen commercial success in the market place are Proban^®^ (hydroxymethylphosphonium salts) and Pyrovatex^®^ (N-methylol phosphonopropionamide derivatives) [7,9,10,11]. The former is defined as tightly regulated treatment of cellulosic fabrics with a pre-polymer of tetrakis-(hydroxymethyl) phosphonium salt (THP) and urea, followed by ammoniation with a mixture of THP-urea (precondensate)-ammonia covered by trademarks [12]. The latter employs molecular fixation using THP and urea as well, with the (CH_3_)_2_P(O)CH_2_CH_2_C(O)NHCH_2_OH chemical unit described in the 1971 patent literature [13].

Despite these early developments in fire safe cloth treatments, HFR additive technologies flooded the fabrics, furniture, textile, and commodity markets. The decline in HFR use—and the precipitating factor for this review—came about by various environmental and health rulings which began identifying carcinogenic, POP (persistent organic pollutants), and toxic processing or burning by-products of these additives [14]. A brief history in the USA can be traced from 2003 when California and Michigan followed EU precedents to limit PBDE (polybrominated diphenyl ethers) mass loadings to present day bans (in the USA, EU, Japan—totaling 22 countries), and industry ceased production [5,15]. Yet, from an engineering perspective at times the material properties and societal needs may justify *some* lenience towards acceptable, limited use of HFRs. To be clear on the risks taken here is critical. HFRs are now ubiquitous chemicals in the environment through their prevalence in domestic goods, commercial products, and, especially, electronic (e-)waste; frequently these objects contribute to HFR contamination—during manufacturing, landfill (end of life), recycling processes, and open burning (particularly e-waste)—of both wildlife and humans causing endocrine (including thyroid and pituitary) disruption, immunotoxicity, fecundity disruption, fetal-child-adolescent development, neurologic function, mutagenicity, and cancer [14,15]. Indeed, the hyper-toxic dioxin molecule is implicated to be responsible for the increased cancer risk of first responders, fire-fighters, and victims to modern fires [15]. As may be the argument with PBDE still in use and passing particular Californian regulations, the performance of the HFR additives mitigates damage to life and property. When evaluated at the holistic level of a complete fire event, HFRs are orders of magnitude less effective at pre-fire mitigation then PFP or AFP (passive or active fire protection measures); HFRs do not reduce the main threats to life—CO emission, irritant gases, soot, and toxic fumes—but rather increase these and add lethal by-products; in conclusion, incorporation of HFRs into products for the home and casual human use appears indefensible [14,15]. For a complete and thorough derivation of the myriad scientifically proven reasons for the discontinuation of current flame retardant policies, the reader is highly encouraged to read EHHI’s (Environment and Human Health, Inc.) comprehensive output [5,15]. A caveat, however, is that specialized applications may still warrant their use where clean or green technologies do not meet the needs—but no longer should human health and environmental impact be neglected as the cost of products and processes, nor should the vast knowledge of these chemicals be discard for their as-of-yet undiscovered *beneficial* applications.

There is also vast scientific knowledge on the differential systems of a fire. Typical cellulose (cotton fabric) fires, per ISO 834, are approximated to rise slowly to 927 °C after 60 min; hydrocarbon fires, including polymers, rise expediently to 100 °C in just 4 min—per UL 1709; and jet fires increase to 1350° in 5 min, as in ISO 22899 [5]. In conjunction with this increase in knowledge of the mechanisms of combustion types, different chemical, physical, and bulk effects have been investigated. The following is an overview—not comprehensive—of the variety of unique properties of some additives and their combinations. 

Camino et al. thoroughly analyzed the performance of APP (with pentaerythritol; PER). For practical purposes, APP is used as a benchmark of the progress that is achievable in reduced toxicity fire retardants. Through simultaneous TGA-DTG analysis, the three steps of APP degradation were demonstrated, the independent step-wise release of both ammonia and water, and mixture ratios greatly influenced the gaseous release rates and temperatures—thus affecting the foaming intumescence [16]. From this study the three APP to one PER ratio was analyzed at 30 wt % for PP (polypropylene); they drew a major conclusion as to the oxidant-independent “condensed” or “solid” phase action of the additive system, which was shown to be both more effective, per weight addition, and safer than halogenated systems [17]. Urea was also concluded to not have synergistic effects within the PP system [18]. Using IR, NMR, EGA, and TGA, the group was able to identify release patterns, correlated to ratio mixtures and loadings, for inert gases, and the particular esterification, cyclization, and corollary mechanisms by which char and gaseous products are formed with the temperature ranges for effective gas release and intumescent behavior for the PP system using APP-PER [19,20,21].

Whether additive or reactive [8], flame retardants were developed with five characteristic effects designed to impart flame retardancy: heat adsorption via endothermic decomposition releasing inert products; enhanced heat or mass flux to protect virgin material; free radical scavenging products to quench the thermodynamic or chemical combustion; decreasing the heat flux or mass transfer across the burning surface via protective layer formation; and modification of the chemical pathway of polymer degradation—see Figure 1 [7,8]. The “Fire Triangle” demonstrates the interdependent components of a self-sustaining flame; any of these components are targets of the preceding characteristics. To evaluate these five FR characteristics, the influencing factors that impact the self-sustaining flame must also be defined: O_2_ content, net heat flux, effective pressure, and char residue formation mechanisms [7]. Inorganic fillers (metal hydroxides, metal oxides, hydroxy-carbonates, borates, silicon, silicates, (nano-) clays, graphene, etc.), HFRs (both additive and reactive), organic additives (particularly containing elemental or conjugated P and-or N), and intumescent systems have been evaluated for a variety of polymers [8,22]. Exemplary in the development of fundamental knowledge, the chemical and thermodynamic mechanisms of halogenated compounds coupled with metal hydroxides and select additives polymer interactions have been well elaborated [7,8]. Figure 2 shows an example system’s mechanism wherein Sb, Cl, and/or Br synergistically improves fire retardancy. 

Significant progress has precipitated from these fundamental understandings of the chemistries, physical reactions, mechanisms, and processing. Some important notes and caveats have emerged. Particularly: the idea that thermal stability does not guarantee increased fire performance; the trend of decreasing TTI (time to ignition) with fire retardant systems as a trade-off for sensitized ignition and increased charring residues; surface migration theories for nanoparticles (convection effects from temperature gradients or viscosity gradients, lower surface free energy of clays and particles, or rising bubbles); and char formation enhancements [10]. Correlation studies have also emerged to better comprehend the combined qualitative and quantitative results of LOI, UL-94, and CC experiments [10]. Synergism has also arisen as a widely-explored method to reduce the costs and environmental concerns of flame retarded products. The components can be traditional FRs, such as those discussed in [5], and, as an example, those explored in [23]; or more progressive, as is evident in Wang et al.’s continuing investigations into bio-sourced organic modifications of APP [24]. Such efforts, however [25], should be more concerned with the propagation synergism of PBDE and associated compounds, in continuing applications where BEST-FR would be more societally beneficial, and focus on applications where these chemicals impart properties not (yet) obtainable or processable from more benign compounds, chemistries, and technologies. These efforts can be observed as proactive efforts. As discussed earlier, the origins of FR trends lie within fabric treatments but proceeded towards bulk material additives; currently, the meta-scopic cycle has returned to a penchant in literature for fabric FR methods. Alongi et al. have been prolific in their investigations in this area. Their recent overview of fabric FR treatments and compounds highlights the vastly advanced technologies now entrained on this issue. Nanoparticle adsorption, layer-by-layer (LbL) deposition, sol-gel, dual curation, plasma treatments, and even biomacromolecules have been investigated in the last two decades, emphasizing the micro and macro-architecture effects on flame retardance efficiency, thereby greatly lowering the wt % quantities necessary [9]. Similarly, legislative action had prompted a return to bio-sourced materials for fire protection in everything from bulk commodity materials to structural element protection—however, more stringent environmental and mechanical tolerances are imposed in current research and development. In the prior area of research, melamine (MA) rose to prominence as an affordable, abundant, and effective intumescent formulation component coupled with APP [23]; in the latter category the removal of asbestos applications from the market has prompted the development of new intumescent paints, coatings, and PFP systems—interestingly even bio-waste such as chicken-egg shells have proven effective [26]. Therefore, some are embracing biologically-derived components which are also greatly supplemented by biomimetic approaches to creating functional architectures. Nacre has been synthetically approximated, and as expected, exhibits phenomenal mechanical strength (Young’s modulus ≈21–34 GPa), and incredible thermal-fire resistance, even at just 50 μm [27]. These last (passing) examples are but a preface of the bulk content of this article, yet illustrate the growth in green, clean, and conscientious chemistry. 

### 1.2. Discovery of Bio-Sourced Fire Retardants

This area shall evaluate the current state of known, reported sources and materials that display inherent natural flame retardance and occur in the natural state. Examples include natural structure, fauna, flora, and bio-mimetic chemicals. 

Living beings that typically endure or thrive in extreme high and low temperatures are termed extremophiles. For this study the latter will be ignored, and focus will be given to species that excel in high temperature environments. There are few animals that meet the requisites for such endurance. Of particular interest are *Thermus* genus worms inhabiting the deep-sea thermal vent habitat. *T. aquaticus*, *T. antranikianii*, *T. igniterrae*, and *Alvinalla pompejana* retain life through the employment of eurythermal life-sustaining enzymes, proteins, and other biological components. Notably *T*_aq_ polymerase as an enzyme is an example. Via this enzyme, *A. pompejana* is able to sustain a symbiotic filamentous bacterial thermal-protective blanket of about 1 cm thickness. Without this layer of protection, it is likely the highly toxic environment of pure geothermal exhaust would slay the worm [28,29]. Albeit (at the time of this writing) no literature can be found related to these fauna-derived products, the biomimetic approach to synthetic materials has revealed molecules such as DNA and chitosan, among other proteins and biomolecules, to be excellent BEST-FRs. 

While plant life seems to have eschewed the natural selection processes necessary for such environmental conditions, there are remarkable life-cycle integrations eminent in the plant kingdom. There is temperature-specific resin expansion in pinecones, redwoods, and chaparral woody plants for seed release or bark cleansing via heat and smoke. Additionally, a subsection of plants has adapted to particularly fire-prone zones with reduced fire risk via spatial or moisture techniques for self-preservation. For example, an extensive systems approach to understanding the best planning for a fire-risk diminishing landscaping via selective planting of certain species has be done. Prominent in the report are: “capeweed, lippia, ice plants, or prostrate myoporum; woody ground covers, such as coyote brush, prostrate ceanothus, and manzanita species; hedges, such as oleander and myoporum; and even some coniferous trees, such as the Canary Island pine and Chir pine” [30,31]. Between these two resources a protective natural and green landscape can be constructed from groundcover, perennials, shrubs (deciduous or evergreen), and trees (deciduous or coniferous). It has been found that in species such as these the lignin, molecular P and N, tannins, and (metabolic) ammonium polyphosphates greatly influence the flammability, combustibility, char formation rates, and decomposition (natural decay) [32]. There does exist a detailed analysis of plant fiber thermal decomposition. Thermo-oxidative studies reveal a three-step process: 300 to 400 °C yields aliphatic char and volatiles; at 400 to 800 °C aliphatic char aromatizes, yielding aromatics, CO, and CO_2_ as carbonization and oxidation co-occur; and, lastly, near 800 °C oxidation leaves just CO and CO_2_ [9,33]. 

Before elaborating on the bio-polymer, bio-fillers, and BEST-FR additives below, specific mention of some co-developing technologies that will ultimately enable their successful integration into existing manufacturing and processing facilities should be mentioned. Ionic liquids have proven to be substantial aids to processing difficult (oil-based) polymers, chitin-cellulose mixtures, and composites. 1-allyl-3-methylimidazolium bromide (AmimBr) and 1-butyl-3-methylimidazoliumchloride (BmimCl), 1-butyl-3-methylimidazolium acetate (BmimAc), and 1-ethyl-3-methylimidazolium acetate (EmimAc) have been used to successfully dissolve and process chitin-cellulose; chitosan-cellulose; starch; and cellulose and rice starch and agar and zein—respectively [31]; this serves as but a stop-gap measure to warn nay-sayers of the processability of bio-derived chemicals that the technology is developing and applicable. A prime example of where bio-materials and green chemistry are coming to the forefront of technologic development is in formate derivatives. Using carbohydrates from waste streams, formic acid for future fuel cells and ammonium formate (used in various industries) can be created [34].

Bio-sourced FR agents in many/most applications will likely need to be supplemented by incorporation of inorganic materials (for example clays, metal hydrates and silsesquioxanes) and/or carbon allotropes, such as graphene or carbon nanotubes. The focus of this review is on the biological contributors to flame retardancy in polymers, so the inorganic and carbon allotropes will not be discussed herein.

The present review intends to build upon a growing body of work, including an excellent recent review by Hobbs [35], which focuses on incorporation of tannic acid, phytic acid, isosorbide, diphenolic acid, DNA, lignin, and cyclodextrin as specific additives to polymers, whereas herein we present more of the historical technology of flame retardants as well the current state-of-the-art for bio-based additives grouped in broader material classes.

## 2. Materials

An outline of the materials to be discussed below is provided in Scheme 1.

### 2.1. Proteins

There are several known examples of naturally fire-resistant biological structures that employ the intrinsic composition or hierarchal structure of proteins and gelatins (Scheme 2). Protein decomposition beginning circa 100 °C gives rise to carboxylic acid functionalized amino acid derivatives and oligopeptide decomposites that catalyze fiber decomposition (particularly cellulose) and lead to higher char yields at high temperatures (750 °C) [9]. Other natural proteins have revealed unique benefits as FRs, fibers, thermoplastic pre-polymers, or combinations of the latter elements [11].

Chief examples that are gaining ground in the fabrics market are types of wool, such as alpaca wool. Alpaca wool exhibits natural fire resistance due to its macro-architecture (fiber orientation and degree of polymerization) and unique protein content (having S containing α-substituents with cysteine as 10 wt % providing cross-linking of peptide chains) [36,37]. The S (3–4 wt %) and N (15–16 wt %) content affords excellent inert gaseous release around 230 °C with disulfide cleavage and amide group cross-linking [38]. It achieves an average LOI of 25 [38]. Other protein fibers have also been evaluated for their use are fillers. Chicken feathers as nano particles (via enzymatic hydrolysis and ultrasonification), separated quills (rachis) or barbs (feather fiber), and whole, have been tested as polymer composite fibers [38]. The unique honeycomb structure affords a low density of 0.89 g/cm^3^, 90 wt % protein content, and respectable moduli—180 Mpa for barbs and 4.7 GPa for quills [37]. 

Aviculture may also yield another approach to BEST-FR advances through another common bio-molecule—keratin, found in abundance in discarded chicken feathers (DCF) as an additive filler. Keratin can be found in a variety of animal species from mammals to reptiles—and hence avians. Its high N content, aside from its protein structure, lends potential to its FR capabilities. Indeed, its degradation profile is notable with three distinct sections: between 115–165 °C only 13 wt % loss is observed; from 255–320 °C another 9 wt % is noted; and finally, between 895–988 °C 25 wt % is observed—for a total of 47 wt % loss—leading to a 40 wt % residue at 1000 °C. Its use in conjunction with melamine and sodium polyphosphate (SPP) with glyoxal as cross linking agent was explored to create a dip-coating intumescent system, initially at ≈6 wt % and 8 wt % (with synergistic 0.026 M borax and 0.97 M boric acid) [39]. Processing steps to create an extracted keratin powder from the DCF are necessary through basic cleavage of the disulfide and peptide bonds, and then precipitation at pH 4, the isoelectric point of keratin [40]. The addition (1:8:5 weight ratio, respectively) of SPP (source of P) and melamine (additional N source and spumific) was chosen to elevate the elemental content of the mixtures; moreover, further processing the mixture above with glyoxal (1:0.018) at pH 5, and obtaining the final product at pH 8 allows for the keratin’s amino, hydroxyl, and sulfhydryl (from earlier cleavage) groups to react with the SPP and the amino-aldehyde groups present within melamine; the latter pairing also serves as the crosslinking points for glyoxal [39]. Not only were both ≈6 wt % and 8 wt % shown to increase the LOI of treated cotton by 67% and 122%, respectively, but a critical loading study was also performed to understand the minimum needed for the synergistic system. Apparently, 5.6 wt % marks the beginning of instant self-extinguishment, with progressively lower burn lengths and afterglow times up to 8 wt %. SEM imaging confirms that the synergistic effect of the borax and boric acid treatments during dipping enhance dispersion onto the cotton surface, thereby creating a thicker, more resilient intumescent char layer during burning [38], likely an emergent contribution of the overlapping degradation peaks of the corresponding additives: DCF (processed) at 290–329 °C, borax at 310–340 °C, and boric acid at 297–331 °C [39].

Chicken eggshell (CES) is an incredible bio-waste from aviculture at 150,000 tons per year (USA) whose disposal constitutes an environmental hazard where production is large [26]. Its composition, among three areas—the shell (cuticle), inner (lamellar), and outer (spongy) membranes—is 95% inorganic calcium carbonate with a mix of collagen type X, sulfated polysaccharides, and proteins composing the sum [5,26]. Indeed, just 2.5–5 wt % of silica fume (SF) (an industrial inorganic waste product) lends increased adhesion to (acrylic) intumescents for steel with improved flame spread and oxidation resistance, and an increase thermal stability in the range of 700 °C due to non-combustible gaseous release which promotes the intumescent system as a spumific [5,26]. The thermal degradation of CES was found to be a bi-stage mechanism; the first from 300 to 790 °C is the 7 wt % dehydration, and the latter between 790 to 907 °C is the oxidation of the calcium carbonate and the release of CO_2_, resulting in a total 53 wt % residue [26]. There is an interesting-weight gain phenomenon for CES depicted in Figure 3 and Figure 4 ascribed to a re-carbonization. An APP/melamine/PER/SF/CES (37:18.5:18.5:7.4:18.5) intumescent system (with 40 wt % residue at 750 °C) was tested on structural steel samples [26]. This coating exhibited exceptionally dense, non-cracking multicellular (expanded foam) char expansion, compared to non-CES systems, and good adhesion contributing to a thermal protection gradient difference compared to bare steel of 450 °C—exceeding qualifying peak temperature tolerance per Eurocode by ≈66% [26].

Whey proteins are effective due to their β-lactoglobulin, α-lactalbumin, bovine serum albumin, and immunoglobulin with high sulfur content in methionine or cysteine residues; these proteins can be utilized in either concentrate (≈53% protein), isolate (≈90% protein), or hydrolysate (≈72% protein) [10,11]. Whey protein isolate (WPI) is composed of α-lactalbumin and β-lactoglobulin and can be collected en-masse from the cheese industry as a waste product [36,41]. Interest in WPI as a FR evolved from its high mechanical properties and barrier properties against oxygen/moisture [39]. WPI in either the folded or unfolded state at 20 wt % addition creates a sensitization of the cellulose (cotton) degradation, conferring an ultimate boon to stable high-weight percent residue and a 33% reduction in burn rate [9]. There is a noted difference in morphologies, however, between the protein states as the unfolded variety incurs tertiary to quaternary flexibility to the WPI FR film [41]. Nevertheless, the WPI cotton treatment residues are stable until complete degradation at 490 °C and 510 °C for folded and unfolded WPI respectively [41]. Folded WPI forms a 30% weight residue while unfolded WPI forms just 5%, yet both inhibit the burn rate by 50% via barrier properties and additional water vapor release [41]. Figure 5 and Figure 6 show the structure of underlying cotton fabric is protected—at an average 500 °C [41]. 

Caseins and hydrophobins—both P and S containing compounds—have been explored as FRs. Unique for these molecules is their ability to adhere to myriad substrates via spontaneous self-assembly of patterned 8-cysteine residues forming non-repeating di-sulfide bonds for hydrophobins (with a similar process for the phosphate-rich groups of casein) in tertiary structures [11]. For protecting substrates such as cellulose, both can be employed to restrict the degradation mechanism toward pyrolysis and preventing depolymerization—thereby producing more char and less volatiles, such as levoglucosan and furan derivates [36].

Casein’s most prevalent components include (but are not limited to) α_s1_-casein and α_s2_-casein, which are the major parts with seryl-phosphates groups around 8–10 in number (β-casein and κ-casein contain about five phosphoserine residues) [10,36]. Casein is recovered from the dairy industry as a by-product of skimming milk. When protecting cellulose, these produce poly-amino acids (phosphoric or sulfidic acids) and oligopeptides endowed with carboxylic acid groups at around 100 °C which catalyze thermally stable char around 570 °C. Furthermore, they extend the decomposition of cellulose into a third—unprecedented in pure cotton—stage circa 602 °C [11]. Caseins exhibit decomposition features which are almost analogous to APP salts [9]. Table 1 shows the decreased burn rate (–35%) and increased residue—a 40% increase in total burn time and −27% decrease in HRR (heat release rate) was also reported—and Figure 7 and Figure 8 demonstrate this ability graphically as globular P rich domains burst during combustion [9,37]. However, the ignition is sensitized, a superficial detriment leading to combustion occurring sooner—yet, for longer and with lower (−27 % versus pure cotton) PHRR (peak-HRR) [36]; this has also been attempted on PET and PET–cotton blends at 20 wt % loadings. The sensitization and phosphoric acid promoted char formation was also observed; PET fabrics saw flame blocking under 30 mm burnt (due to an increased LOI of 26%) with an overall −67% BR and thus a total residue of 77%—with virgin material; meanwhile PET–cotton fabric blends also experienced flame retarding properties. They were not as marked as the PET-only fabrics’—but exhibited no dripping, as the PET did; see Table 2. Casein has also been shown to be an effective flame retardant when incorporated into poly(lactic acid) [42].

Gelatins, derived from bovine or fish sources, have been shown to reduce flammability in some polymer systems, including poly(vinyl alcohol) and polyolefins [43,44]. The mechanism of this FR action has not been thoroughly investigated to present, but may include a combination of crosslinking, which limits dripping and allows for char formation, as well as the generation of carbon dioxide and other non-flammable gases.

Hydrophobins emerge from filamentous fungi rich in cysteine containing proteins (circa 7–9 kDa molecular mass)—comparable to the content found in keratins of wool [10,36]. These are qualified, despite their characteristic octo-cystiene bridge scaffold structure, by differences in hydrophobic and hydrophilic amino-acid residue distributions, into two classes: I which are insoluble in aqueous media and have low wettability, with SC3 and EAS as leaders; and II, which aggregate but are water soluble, with HFBI and HFBII as eminent members [10,11,36]. Beginning its degradation circa 200–270 °C, releasing phosphoric and sulfidic acid (and gaseous H_2_S) as disulfide bonds break, it promotes a stable char around 570 °C. Furthermore, a sample dipped in mixed hydrophobins extended the degradation of cellulose (with a 20 wt % coating) beyond the pure cotton into a (an unprecedented) third stage circa 620 °C [36]. Table 3 shows the BR reduction and residue increase, while Figure 9 vividly demonstrates this action through spherical domains which protect the underlying fabric [36]. The ignition of the treated fabrics is sensitized; however, the BR is reduced and the PHRR is 45% decreased compared to pure cotton [9,36].

Corn, albeit mostly a source of carbohydrates in the diet, yields zein from its endosperm as waste products of both agriculture and bio-fuel production (especially ethanol) [45]. Zein is viable as an FR due to its high N content within glutamine (20–26 wt %), leucine (21 wt %), proline (10 wt %), and alanine (10 wt %); this mix of amino acids also yields a 50–50 distribution of polar-non-polar bonds lending zein a unique amphiphilic characteristic. This ratio is indicative of the content within the zein; however, 80% of the non-polar prolamine is contained within the α-zein “half,” leaving β-zein as the aqueous-compatible component. Processing concerns for corn zein have been well explored with heat and shear, enabling compression molding, melt mixing, extrusion, and foaming of thermoplastic zein (TPZ) [45]. TPZ is enabled by the addition of alkoxysilane compounds (especially POSS—polyhedral oligomeric silsesquioxanes). In one study a sol–gel-prepared neat TPZ was compared to 25 wt % polyethylene glycol and 1 wt %, 1.5 wt %, and 3 wt % final silane content 3-glycidoxypropyltrimethoxysilane (GOTMS, an epoxy functionalized POSS, selected to provide zein amine and GOTMS epoxy hydrolysis and condensation [45]. The hydration of the resultant TPZ-POSS systems exhibit exceptional tensile moduli (≈5, 9, and 11 Mpa, respectively with increasing GOTMS) and acceptable elongations at break (60%, 83%, and 106%, respectively). 

A similar molecule to zein in origin, gluten (WG, containing 77 wt % protein (50 wt % glutenin, a poly-peptide, and 50 wt % gliadin, an aminoic acid), 6 wt % starches, and 1 wt % lipids) can be obtained as waste from the bio-fuel (ethanol) production from wheat. Reacting the WG with TEOS first in basic (pH 11) and then at elevated temperature (90 °C) during the foaming process yields a foam with molecularly incorporated silica [46]. The foams were examined via three variables for flame retardance and mechanical properties: silica content (1.5, 3.1, 6.7, 11.0, 16.1, 22.4, and 53.6 wt %) per TEOS loadings of 5, 10, 20, 30, 40, 50, and 80 wt %; fast N_l_ foaming (−196 °C) and slow CO_2_ foaming (−25 °C); and cross-linking agent glutaraldehyde at 8 wt % or 16 wt %. Samples either achieved V-0 qualification per UL-94 (CO_2_ samples containing 11.0, 16.1, and 22.4 wt % silica with 8 wt % glutaraldehyde; and N_l_ samples with 16.1, 22.4 wt %) with under 1 s self-extinguishment, or failed—independent of cell size; i.e., foaming temperature. The effective concentration differential between 6.7 and 11.0 wt % silica marks the transition between a decreased mass loss rate, formation of inorganic silica layers, and thinning cell walls—with no change in the presence of amide (I or II) nor S products within the representative chars. However, the N_l_ samples which all exhibited cross-linking with the gliadin, and the higher content Si foams, showed substantially worse mechanical properties due to thinning walls and uneven structures—this problem was ameliorated satisfactorily in the CO_2_ systems with the addition of the glutaraldehyde [46]. 

### 2.2. Amino Acids and Oils

Development of polymeric materials from amino acids directly, instead of their oligomeric and polymeric compositions, afford calibration of the thermal, mechanical, structural, and solvent response characteristics (Scheme 3). An example of where these technologies could be particularly useful is in medical application materials. Levo-phenylalanine-based (with urea) materials can significantly out-perform PLA [47]. Linear, branched, and hyper branched monomers and fibers of these, can be created from the same, with variations of alkyne, azide, alkene, amino-acid, carboxylic acids, and ketones (etc.) making these monomers compatible with myriad chemistries. Likewise, development of polymers from fatty acids, such as stearic acid, can be free-radical homo- or co-polymerized with nano-scale additives to create polymers from bio-sourced materials [48].

The once rarified DNA molecule is now an eminent molecule for fire retardancy following its easy isolation from both salmon milt (herring sperm) and roe sacs—both waste products [10,49]. DNA functions as a single-biomolecule intumescent system containing all prescribed components of an effective intumescent system. Phosphate groups are the acid source via phosphoric acid; deoxyribose rings decompose acting as both carbonifics and spumifics via char producing dehydration; and the bases (e.g., guanine, adenine, thymine, and cytosine) release ammonia as another spumific or gaseous dilutent [9,10,11]. DNA has been shown to also resemble APP in its thermo-oxidative performance with an initial decomposition releasing phosphoric acid (or chemically similar units) circa 200 °C, thereby promoting a stable char above 500 °C [9,49]. Following ASTM 3659 a 19 wt % addition of DNA prevents burning from the 3 cm methane flame in the horizontal figure yielding 98% virgin residue [9,50]. The series was expanded to cover 5 wt % and 10 wt % additions, concluding that despite the weight percent of DNA vastly increasing the thermal stability and forming a consistent residue with 10 wt % being the limit for effectiveness obtaining self-extinguishment—19 wt % provides the same limit for “no ignition” in the 35 kW/m^2^ CC experiment—with both decreasing the pHRR (peak heat release rate) by 50% [9]. Despite these initially encouraging results, increasing the heat flux to 50 kW/m^2^ greatly reduces the fire retardancy of this method of solution application. 

Layer-by-Layer (LbL) is a manner of nanoparticle chemistry emergent in 1991 primarily using electrostatic interactions to fixate thin layers via polyanion-polycation couples into multilayered, and at times multicomponent, coatings on a substrate [9]. This methodology is advantageous for myriad scalable benefits: dilute, environmentally benign colloid or polyelectrolyte solutions; possible automation; a variety of application techniques (spray-on, dip coating, etc.); controlled, precise application-tailored customization; and low weight addition—just to name a few. 

DNA and chitosan (see next section) have been explored as a combined LbL system in 5 (2.5 wt %), 10 (7 wt %), and 20 (14 wt %) layers via CC and UL-94 testing [9]. The systems performed thusly: five layers—0% reduction in BR, 9% increase in residue (from 2% per CC), 3% increase in LOI (18% base LOI for cellulose), 25% reduced pHRR, 10% reduced THR (total heat release); 10 layers—13% decreased BR, 10% increase in residue, 5% increase in LOI, 38% reduction in pHRR, and a 21% reduction in THR; lastly, the 20-layer system achieved a 30% reduced BR, 11% increase in residue content, 6% increase in LOI, 41% reduction in pHRR, and a 31% reduced THR [6]. For the content of material applied, these results are extremely encouraging for future development. Another bi-layer system of poly(allylamine), obtained from either *Brassica nigra* or *brassica juncea* (black or brown Indian mustard seeds, respectively) and poly-sodium phosphate 10 BL reported substantial decreases in both THR and PHRR [9]. Increased to 20 BL, the same system completely prevented ignition of the treated cotton fabric [49]. An amino-derived poly (acrylic acid) and Na-MMT clay 20 BL system also produced substantial TTI and THR reductions, with a modest PHRR suppression. See Table 4 for a listing of all of these results [9].

Vegetable oils such as those from soybeans, palm trees, linseeds, sunflowers, castors, and olives afford polymer precursors with straight chains of unsaturated double bonds readily polymerized via free radical, cationic, click (azide addition), ring-opening metathesis, or condensation [39]. Acrylic epoxidized soybean oil (AESO) and soybean oil monoglyceride (SOMG) are exemplary starting points for use in co-polymerization with petrochemical-based polymers, thereby commencing their phasing-out. These studies point that the degree of crosslinking can be controlled and lead to development of satisfactory replacement composites. Plant oils and phenolic molecules are also finding use by conversion to FR derivatives, such as their phosphorous esters [51].

### 2.3. Carbohydrates

A range of carbohydrates are found in nature, and a number of them have been found to be useful components in flame retardant compositions, Scheme 4.

Plant fibers are not thermoplastic in behavior—characterized by a pyrolysis or decomposition temperature, *T*_d_, below their glass transition, *T*_g_, and/or melting temperature (if crystallinity is sufficient), *T*_M_; they are mainly composed of cellulose, hemicellulose, lignin, and a remainder mixture of varying pectins, waxes, and inorganic or organic compounds [33,37,52,53]. Cellulosic fibers (being the most common in extant use) include cotton, jute, flax, sisal, kenaf, hemp, ramie, abaca, oil palm, sugarcane bagasse, bamboo, pineapple leaf, coir, date palm leaf, curaua, rice straw, wheat straw, and cornhusk fibers—all of which have been investigated as fillers. Table 5 gives a good illustration of the variety of fiber compositions. Table 6 shows the different classifying groups for these fibers and their differing contents of cellulose, hemi-cellulose, lignin, pectin, wax, and ash [54]. A number of key processes have been identified during the degradation of these fibers. Chief among them is the catalytic emergence of levoglucosan; accompanied by desorption of water and cellulose intra-molecular bonding and cross-linking, to form dehydrocellulose, volatiles (gaseous and solid), char, and tar [37]. Levoglucosan is widely known to release the greatest number of decomposition molecules—the flammables and volatiles include unsaturated and saturated oligomeric hydrocarbons and combustible small molecules. Cellulose has a broad decomposition range, 260 °C to 350 °C with myriad volatiles, combustible and non-combustible gases, tar, and some char; hemicellulose has a slightly sharper range of 200 °C to 260 °C and releases more non-combustible gases, but less tar [37]. 

Analyzing some of the fibers form Table 6, flax, jute, and sisal have been reported to have similar thermal degradation, per TGA, with 340 °C for both jute and sisal—and flax a slightly higher 445 °C [53]. These are significantly high when compared to Table 7 showing the commodity plastics’ *T*_d_. The low lignin content of flax afforded a higher *T*_d_, but less oxidation resistance comparable to the jute and sisal fibers with high aromatic content. Rice fiber, as another example, has notably high silica (ash producing) content and low content for both hemi and regular cellulose, and so should be more fire resistant than cotton fiber [37]. Industrial hemp fibers within polyester fibers helped reduce the pHRR by 51% through ignition delay and reduction in volatile gas release [54]. Lastly, lignin has a *T*_d_ start of 160 °C but ends near 400 °C for the weaker bonds break earlier while the larger, more stable aromatics have scission at higher temperatures—thereby contributing more char than either of the latter components.

Fiber with increased fire resistance can be crafted by blending novel material resources. Wool fiber waste and cellulose acetate were combined to create a keratin-cellulose filament with superb fire resistance. The structure of the fiber, and the diameter, length, aspect ratio, and angle between axes are also critical [37,54]. The orientation of the fiber (as related above by alpaca wool), the polymerization degree, and the crystallinity play important roles. Higher cellulose crystals give rise to more levoglucosan during decomposition, but afford higher ignition temperature due to the latent heat of activation necessary to melt the crystalline domains [37]. This is exemplary in amorphous cellulose requiring 120 kJ/mol compared to crystalline cellulose (from cotton and ramie fibers) needing 200 kJ/mol. Fibers that are longer, larger (by diameter), structured (woven, matted, etc.), and aligned in the direction of loading all have improved mechanical strength. Both the polymerization degree and orientation either facilitate or inhibit the oxygen diffusion during decomposition. When comparing cotton and ramie, ramie’s high orientation affords a higher thermal stability limit. The effect of these two components can be demonstrated best by the extrapolated interaction of their structures in blended composites. An effect observed both in mixed-fiber composites and due to weight percent addition, in matrix systems, is the reinforcement of molten matrix products’ proximal to and contacting the heat source—this is termed “scaffolding effect” [37]. 

Lignin itself, due to its composition of hydroxyl and phenolic hydroxylated groups, can be made into an epoxy resin whether reacted with epichlorohydrin (ECH) or chemically treated to functionally behave as ECH [55]. Ferdosian et al. report that the de-polymerized (by either Kraft or organosolv methods) lignin reacted with 4,4′-diaminodeiphenylmethane (DDM) yields epoxies that are highly competitive compared to traditional BPA epoxies [56]. Albeit the LOI was derived using Van Krevelen’s equation, those values are still impressive. As a general trend, from TGA analysis, increasing lignin content decreases the activation energy for initial (first step) decomposition-degradation, but later emphatically increases the activation energy for the second step [55]. Two DDM and “Kraft” lignin-based epoxies with contents of 75 wt % and 100 wt %, and one DDM and “organosolv” lignin-based epoxies (also 100 wt %) reached respective LOI values of 30.3, 32.7, and 29.1. Noteworthy, is that—per Van Krevelen’s equation—none of the lignin based epoxies of the study performed below the level of BPA based epoxies [55]. Indeed, glycerol, sorbitol, isosorbide, gallic acid, furan, and itaconic acid can be organically obtained and used as epoxies precursors. Since these exemplify advantages of poly-phenolic structures, they are covered in depth in that section (poly-phenols) [57,58]. 

Bourbigot et al. have demonstrated that 20 wt % pure lignin can reduce the pHRR for various oil-based polymers (PP, PU, and PBS) by a minimum of 30% [58,59]. Hydroxylation, alkylation, amination, nitration, and phosphorylation techniques can be utilized to further enhance lignin’s effects. Phosphorylated lignin will be used as an example of these techniques. After treatment whereby 4.1 wt % is added, the degradation onset of lignin is decreased to 215 °C as a tradeoff for enhanced thermal stability (60% reduction in mass low rate—MLR) and increased char yield after 400 °C. At a high loading of 30 wt % in ABS, the phosphorylated lignin sensitizes the degradation onset by 20 °C (down from 325 °C) but reduces MLR by 25% and increases total char yield by 17% [58]. The decrease in degradation onset is replicated in a lowered TTI (50s from 80s), yet pure lignin reduces the pHRR by 43% and THR by 13%, and phosphorylated lignin reduces pHRR by 58% and THR by 20%, demonstrating the change in emissivity attained with addition of lignin products—especially with the introduction of just 1 wt % P. The gaseous release of dangerous products is also reduced, as phosphorylated lignin in ABS evolved 23% less CO_2_ and 29% less NO—both irritant and deadly gases in a fire scenario—while maintaining water release and increasing cyanuric acid release by 17%. The authors concluded that the treated lignin modifies the decomposition mechanisms of ABS, leading to more incomplete combustion. These results demonstrate that with little more effort, the treatment of lignin could yield substantially more effective FR additives. 

Above entails a “bio-source” approach, which is not the only way to obtain utile lignin. Lignin is a by-product of the paper-making industries yielding some 50 million tons of alkali-lignin, which is typically classified as waste [52]. The same principles of (wheat straw) lignin treatment are also employed here to purify the molecule and then increase the N and P content within the final matrix, PP with wood-product (WP, wood pulp containing hemi-cellulose, cellulose, and pure lignin) filler, via grafting [52]. A functionalizing of lignin is shown to be possible via polyethylenimine (PEI; contributing N) and diethyl phosphite (DEP, contributing P, with Cu^2+^ included from processing) as lignin grafts [52]. A curious mix of results are obtained for the loading series studied: 15 wt % PEI-lignin additive and 10 wt % DEP-lignin achieve V-2 rating, while 15 wt % DEP-lignin obtains V-1 rating; these qualitative results are contrasted by the CC quantitative results showing just 5 wt % DEP-lignin demonstrates the highest (36%) drop in THR versus the neat matrix of 93.2 MJ/m^2^. It is concluded, from literature analysis, that the optimal residual (0.33 wt %) Cu^2+^ content catalyzes effective char formation for the DEP-lignin system, but above this level the ion weakens the char structure formed. Both DEP-lignin and PEI-lignin decrease the TSR and SPR as well; however, the differences in char structures, studied by RAMAN spectroscopy, assure that the graphitization of the DEP-lignin/PP-WP is greatly aided by the Cu^2+^, Cu^+^ ions, and reacted Cu present in the char remains, and is furthered by the P-derived acid species’ preferential decomposition of organic products. Thus, the synergistic effects of the DEP-lignin complex outperform—at low wt %—the PEI-lignin systems, and both complexes see decreased performance with increased wt % [52]. 

An encouraging start considering composite blends can see synergistic effects with other additives. For example, polyurethane with either corn shell or wood flake as a bio-filler with APP saw increases in the reported LOI [37]. Sugarcane bagasse waste can be treated to be used as an inexpensive fiber filler with adsorptive properties for both water- and fire-proofing agents—this highlights the potential for obtaining more synergistic composite components, thereby reducing the loadings needed to impart sufficient flame retardation [52].

Alginate (AG) is natural carbohydrate salt (of alginic acid) found within various algae species (red or brown) and seaweed [6]; it can be extracted pure as an anionic, random block copolymer of β-1, 4-*D*-mannuronic acid (M) and gluronic acid (also known as α-1, 4-*L*-guluronate; G) within regions of MM, GG, and GM/MG units [6,52]. This structure is highly malleable with films, fibers, and aerogels processable in facile conditions and at relatively low cost [6,60,61,62,63,64,65,66]. 

The latter material, an aerogel, is created via a freeze-drying methodology—with 5 wt % Na^+^MMT and 5 wt % ammonium-AG, just 5 or 15 wt % ammonium-AG, and/or with various cross-linking ions—of low environmental impact that takes advantage of AG’s LOI of 48 and near-complete combustion characteristics (3 wt % residue and CO_2_ as 97.85% of products) [60]. The ion chosen in this study was Ca^2+^; physical cross-linking at neutral pH may be accomplished as: direct, with addition of 0.25 M solution of CaCl_2_ to AG gel, yielding Ca^2+^ cross-linking; or indirect, with a 0.005 mol CaCO_3_ solution-suspension (from 0.005 mol solutions of Na_2_CO_3_ and CaCl_2_) added to the AG gel with 0.01 mol of gluconolactone (GDL) to generate viable Ca^2+^ cross-linking ions from the stronger Ca^3+^ ions via controlled release via GDL’s action. Mechanically the Na^+^MMT and ammonium-AG content contribute to a stiffening of the aerogel with synergistic effects (for example: the 5/5 wt % system shows 230% increased TTI, non-linear increase in pHRR, prevention of ignition—even under 50 kW/m^2^ CC testing, 566% increase in char residue, 75% reduced FIGRA, and −15% HR/g change) and beyond-density factors as AG concentration transitions from layered to network architectures; and Na^+^MMT creates localized, disordered co-continuous networks—reflected in the viscosity–structure relationship established for aerogels [61]. Hence, increased viscosity due to ion-induced physical cross-linking would again create altered network structures [60]. This ability to tailor the aerogels generates materials with mechanical strengths similar to balsa wood or standard PU foams [61]. Either Ca-ion source doubles the compressive modulus of the aerogel, yet the indirect system provides a seven-fold increase, with an optimal limit between 2% and 3%.

Fibers can be made from AG and Ca-/Ba-/Cu-cation mixtures via wet-spinning [60]. These fibers exhibit values that encourage their uses in myriad applications; as an example Ca^2+^–AG fibers incur values such as, 1.5 dtex, and 2.65 cN/dtex tensile strength, LOI of 34%, 5 kW/m^2^ pHRR, and 0.46MJ/kg EHC [60,61]. Compared to standard cotton fibers, these exhibit phenomenal performance. During the initial thermal decomposition of these Ca^2+^–AG fibers from 200 to 350 °C, the decomposition products are extremely varied—however, between 350 and 600 °C a transition to only CaCO_3_ and Ca(OH)_2_ as products occurs, demonstrating a preference for carbonization incurred by the presence of the Ca^2+^ cations [66]. 

Despite the more obvious implications of flame resilient (fire proof or non-igniting) aerogels (similar to foams) and fibers, most research in AG has focused on films using 5 wt % Na^+^AG and divalent or trivalent metallic cations. The ions reported here are, trivalent ions, include Al^3+^, Cr^3+^, and Fe^3+^; and the divalent ions represented are Ca^2+^, Mn^2+^, Zn^2+^, Cu^2+^, Ba^2+^, Ni^2+^, Sr^2+^, and Co^2+^. A 0.1 mm 5 wt % Na^+^AG film alone has its first decomposition stage from 200 to 290°C with a sharp mass loss rate peak circa 245 °C, and its second from 219 to 453 °C; additionally, a pHRR of 105 W/g, 47 wt % residue and a reported LOI value of 24% [62,63].

A 0.4 mm Al^3+^–AG film also demonstrated two thermal degradations regimes, wherein the first between 135 and 340 °C demonstrated high mass loss rate (down to 52 wt %) and evolved primarily CO_2_ and H_2_O via glycosidic bond cleavage, and the second stage up to 635 °C involved carbonization to leave 30 wt % residue at 800 °C [6]. Overall, the mechanism proposed for Al^3+^–AG films is consistent with the Py–GC–MS observed product distributions—furfural, CO_2_, 2-acetyl oxirange, and 2,3-butanedione—comprising ≈94% of gases, with pathways increasing in probability from A (tautomerization, cyclic hemiacetal, and dehydration reactions) to B (ring opening, decarboxylation, and tautomerization reactions) to C (all of the previous) with increasing temperature as oxidation became a major mechanism circa 222 °C—all contributing to an LOI rating of 42% and UL-94 V-0 qualification [6,62]. A 0.1 mm Fe^3+^–AG film demonstrated a reduction in fire performance versus thicker Al^3+^–AG films, with a lower and broader first decomposition stage from 93 to 290 °C and a lowered residue of only 28 wt %, but restricted oxidation effects on the film until ≈360 °C, decreasing the pHRR (versus Na^+^AG) by 85%, and earning an LOI of 35% and V-0 rating [61,62]. A proposed mechanism for Fe^3+^–AG is shown and is derived from the main degradation products of CO_2_ (41%), furfural (23%), 3-penten-1-ol, 2, 3-butanedione, and 2-cetobutyric acid-methyl ester as 78% of the gases from dehydroxylation, decarboxylation, ring scission, and glycosidic bond cleavage catalyzed by the iron cations, while esterification is suppressed [63]. 

Divalent Ba^2+^ was added to create 5 wt % 2mm AG films with circa 11 wt % cation loading rated V-0 and 52% LOI [60,63]. Compared to the standard 5 wt % Na^+^AG film’s degradation profile, the Ba^2+^ samples demonstrated a decreased first pHHR of 16 W/g (−85% for 185–340 °C) 40 °C above the norm and proximal results for the second pHHR (only −20% difference for 185–340 °C) at the same temperature and yielded a lower residue of 40 wt %. That latter initial degradation peak increase can be attributed to the ordered, coordinated water content of the barium–alginate blend, which was released prior to the first peak—thereby offering convective cooling. The second peak actually preceded a third stage from 470 to 560 °C which further decomposed the decarboxylation, dehydration, and glycosidic-cleavage products from the second stage. From TGA-FTIR and Py–GC–MS analysis the degradation mechanism has been hypothesized in the literature: an increased catalyzation of decarboxylation and esterification to form a char for this. It is worth reporting the major (≈73 wt %) products of its degradation, by group: CO_2_, ketones (1-hydroxypropan-2-one, propan-2-one, 2,3-butandione and butan-2-one) and aldehyde (furfural), and hydrocarbons (toluene and ilk) [60]. Liu et al. have pursued this research to include reports on Co^2+^–, (15.3 wt %) Cu^2+^–, (13 wt %) Ni^2+^–, and Mn^2+^– 0.1 mm 5 wt % AG films [63,64]. Co^2+^–AG films achieved V-0 and 45 LOI, reduced the first pHRR by 86% at an increased (by ≈50°C) 266°C, reduced the second pHRR by 23% at an increased (by ≈60 °C) ≈520 °C, and left a residue of 32 wt % [65]. Its main products (85 wt %) during thermal-oxidation are reported as CO_2_ (31% alone), furfural (20% alone), hexadecanoic acid, butyraldehyde, propan-2-yl tetradecanoate, acetic acid, and benzene which are modified degradation-scheme products, as observed in, of standard Na^+^AG films [65]. Cu^2+^–AG films were rated a 23% LOI and NR; a unique single-peak HRR curve demonstrating a decreased, by 60%, pHRR of 41.1 W/g at 221 °C; and a final residue of 35 wt % [64]. TGA-FTIR and Py–GC–MS analysis demonstrates that among the myriad products obtained CO_2_ (34 wt %) and furfural (32 wt %) form an evenly distributed majority of the gaseous products as 64 wt % and give substance to the primarily fragmentation mechanism proposed which emphasizes the lack of both intra-molecular esterification and decarboxylation with the Cu^2+^ addition. Ni^2+^–Ag films obtained V-0 rating and a LOI of 50%, displayed the common two-peak HRR curve of AG materials with a first pHHR reduced by 88% to 12.1 W/g at 265 °C (35 °C higher) and a second pHRR reduced by 42% to 18.1 W/g at comparable 447 °C; all resulted in a final residue of 32 wt % [64]. These films saw gaseous evolution of CO_2_ (41 wt %), furfural (19 wt %), pentanal, and 2,3-butanedione resolved 74 wt % of all products, which combined with the TGA-FTIR and PY–GC–MS studies suggest a catalyzed decarboxylation and intra-molecular esterification bi-pathway degradation mechanism. Mn^2+^–AG films received NR and an LOI of 31%, reduced the first pHRR by 90% at an elevated (by ≈30 °C) 266 °C, reduced the second pHRR by 57% at circa the same temperature as the standard Na^+^AG film, and resolved after visible smoldering a residue 32 wt % [64]. Under thermo-oxidation, this film system’s main evolutions (72 wt %) are CO_2_ (47%), furfural (12%), butyraldehyde, and acetic acid, which leads to a proposed mechanism. These systems demonstrate the surprising synergistic or catalytic effects (or lack thereof) that are observed, or are to be discovered, within the BEST-FRs. 

Other plant products can also be used to create novel materials with intrinsically beneficial properties and controlled environmental impacts. Thermoplastic starch (TPS) from, but not limited to: wheat, rice, corn, potato, oat, or peas can be manufactured via extrusion, solution casting, injection molding, and compression molding [53]. Albeit TPS alone suffers poor mechanical and chemical resistance, composites with bio-fillers (phyllosilicates, other polysaccharides—or even those mentioned in this report) have vastly increased thermal, mechanical, and chemical resistance [53]. Other materials that follow this trend are polyhyrdoxyalkanoates (PHA) such as poly-(3-hyrdoxybutyrate) (P3HB) and poly-(3-hydroxybutyrate-co-3-hydroxyvalerate) (3HB-co-3HV) obtained from waste waters, bio-waste, and sugar molasses [53]. 

Starches can also be a source for other materials. Cyclodextrins are given here as an example. EVA, Nylon 6-6, Nylon 6, PP, and LLDPE can incorporate nanosponges of cyclodextrin which is itself stable in air up to 300 °C [66]. The structure of β-cyclodextrin (CD) as a truncated cone; of note are the cyclic 6-8 oligosaccharide units bonded via α-1,4 glycosidic bonding further (intra)cross-linked by organic carbonate linkages which all act as carbonifics and spumifics—while still having thermally isolating empty spaces. They contain hydroxyl groups CD sensitized to the onset of degradation for the polymers investigated, while promoting or enhancing char formation with modification of the native thermal decomposition pathways. CD and CD with added triethylphosphate (TEP) or APP were systematically investigated; what follows are the key results—all those reported here are the most efficient results with added TEP or APP content. PP’s decomposition is step wise at 323 and 480 °C, with THR and EHC 9 (effective heat of combustion) reduced by 11% and 6% respectively; LLDPE is also two-step at 406 and 537 °C, but sees increases in THR, pHRR, and EHC and more smoke release; Nylon 6 is sees its first at 450 °C and the second is pushed higher, but smoke release increased and THR reduced—trade-offs [67]. Cyclodextrin has also been shown to coordinate with phenylphosphonic diamide, which formed a successful FR agent complex for epoxy resins [68]. Covalent attachment of a phosphorous FR moiety, in this case phosphate carbamate, to starch, similarly, has been shown to be effective, providing the active component for flame retardation with a convenient carbon source [69]; a similar approach to flame retardation of epoxy resins with phosphorous derivatives of bio-sourced tartaric acid was equally effective [70].

The LbL technique can also be applied to hybrid systems. Chitosan (available deacetylate), an amino-polysaccharide obtained from crustaceans and insects, has a favorable pK_a_ 6–6.5 and has C, N in such configurations as to be a spumific and inert gas release agent [70]. An attempt is reported for 5 BL and 10 BL of Chitosan-APP on 70% cotton–30% polyester fabrics as an intumescent system on the microscale to test the P–N synergism [9,70]. The study of chitosan deposition was closely observed and noted an increased pH effect on the globular nature of the molecule. These simple BL proved effective in reducing the afterglow phenomena, yet quad-layer (QL), chitosan-APP layers combined with silica-chitosan-silica-APP layers, and QL architectures proved to be stabilizing the residues after both VFT (vertical flame test; UL-94) and CC [9]. Removing the inorganic elements from these systems was also investigated; 30 BL (18 wt % to the fabric) of phytic acid (having 28 wt % P, the bio-storage biomolecule within cereals, bean, and oil seed crops) and chitosan decreased the pHRR by 50% for pure cotton fabrics in a μCC (PCFC) test. Phytic acid may also be used in other LbL systems due to its wide pH interaction range—pK_a_ 1.9–9.5 [69]. That 30 BL system, treated at pH 4, self-extinguished and preserved 90% virgin material; similar systems of 16 wt % (pH 5 processing) and 14 wt % (pH 6 processing) addition respectively preserved and 42% of the fabric weight [70]. This demonstrates the processing efficiency of the chitosan system in aqueous solution and a strong dependence on the P content, as phytic acid concentrations diminished in both the pH 5 and 6 systems [70]. SEM analysis also corroborated the efficacy of the system as the cotton fiber structure was not altered by the LbL deposition and globular chitosan domains expanded in intumescent like manner preserving the material and structure underneath post-burning. Despite variations in pH processing (hence P content and layer thickness) all systems evaluated reduced HR by 70% and increased char residue by (at least, if not greater) factor of 7. Chitosan has recently been examined more broadly as an FR additive, working in concert with a broad range of inorganic additives [71] and phosphorous sources [72]. Phytic acid has similarly been shown by multiple workers to be a useful FR agent, working synergistically with tannic acid [73], or as a phytate salt with ammonia [74], metal ions [75], piperidine [76] or 1,6-hexane diamine [77]. Phytic acid is an intriguing additive in that it brings six phosphate groups per molecule, bound to a readily carbonizable glucose ring; many of the reported salts introduce nitrogen in their cations.

### 2.4. Polyphenols/Polyhydroxyphenols (Antioxidants)

PBDE and its kin have been in phase-out or voluntary cease-of-production since the early 2000′s per (EU) Directives 2003/11/EC, 2002/95/EC, and a mandate by the European Court of Justice (09/05/2009)—reciprocated by changes in USA regulations [78]. Early adopters to the new market amended “organo”-phosphorous flame retardants, primarily resorcinol bis (diphenyl phosphate) (PBDPP or RDP in literature; the latter in this report; used in ABS, PC, PC/ABS, and PPO/HIPS matrix systems) and bisphenol A bis (diphenyl phosphate) (BPA-BDPP or BDP in literature; the latter in this report; used in HIPS, PC, PPO, and PC/ABS matrices) [78]. These are important first steps in the replacement of harmful FRs, but care should be taken to not repeat the mistakes of the past (Scheme 5). 

Phenolic compounds have inherent flame retardant capabilities due to their conjugated structures; the BEST-FRs are summarized below as being highly effective. However, phenolic compounds are already listed as “priority pollutants” by the EPA (USA; for use in myriad industries) through waste-water disposal—resorcinol, catechol, pyrogallol, hydroquinone, and phenol proper are thus worrisome materials for BEST-FR use [79]. Lobo et al. investigated the aerobic oxidation, an analogous process for natural bio-accumulation (pH bio-degradation; i.e., the Haldane equation via respirometry) and some phases of the flame retardant action-mechanism, for the latter compounds. All compounds, save catechol, which exhibited a 50% increase in oxidization coefficient (*Y*_0/S_, per mole O_2_) with increasing concentration, displayed constant *Y*_0/S_ values. Phenol, resorcinol, and hydroquinone gave true Haldane dependencies, while pyrogallol and catechol displayed bi-phasic patterns; ultimately the degradation rates and auto-toxicity degradation (inhibition of degradation) trend are reported as: catechol > phenol > pyrogallol and resorcinol > hydroquinone; and pyrogallol >> catechol and resorcinol >> phenol > hydroquinone, respectively [79]. These trends demonstrate that the complexity of a molecule is not indicative of its degradation rate, but rather its own inhibitory products and mechanisms can greatly impede degradation—albeit these results were extended for clean-up facilities, the implications for mechanistic studies of flame-retardant bio- and heat- degradation are apparent. 

Tannins have taken the mantel of bio-sourced poly-phenolic structures in both bio-materials and BEST-FRs. Figure 10 gives the base-units and (general) reaction sites used to generate adhesive and flexible films from mimosa (*Acacia mearnsii*; harvest ≈220,000 ton/yr. tannin extract (80%–82% flavonoid content of which 90% is robinetinidin; average degree polymerization 4–5 and linked C4–C6 [80]) and quebracho (*Schinopsis lorentzii* or *Schinopsis balansae* harvest ≈80,000 ton/yr.; 80% consisting of resorcinol-A and catechol-B rings, the fisetinidin flavonoid unit [80]) via amination with furfuryl alcohol and either glycerol or PEI [81]. This demonstrates facile ways to potentially generate strong, energy dissipative BONDS within bio-sourced materials with tunable transparencies, adhesion, and tensile-ductile behaviors [81]. Bark tannin from the same mimosa family has been explored as a replacement for 10-ply phenol formaldehyde plywood laminates using a similar furfuryl alcohol (in NaOH-pH adjusted water) and “fromurea” (from the self-same plant), termed a tannin/uranic resin [82]. ASTM D 4060-01, EN ISO 2409, EN 438-2, EN 314-1, and a flame test (Bunsen burner flame tip at 7 cm below sample) were performed resolving favorable results (see Table 8 with potential applications for construction, mechanical, and electronic engineering supports or varnishes—albeit this initial formulation demonstrates a 30s (37%) sooner burn-through time [81]. Applying similar formulations better results are readily shown to be achievable. Testing a series of 10–20 wt % tannin in 50 wt % (versus hexamine wt %) NaOH-water (to a corrected—with added 5 wt % boric acid or phosphoric acid—pH 9.0) mixtures with EN 113 selected scots pine (*Pinus sylvestris*) and European beech (*Fagus sylvatica*) at 12% moisture content for fire retardancy (50 × 25 × 15 mm^3^ samples at 2 and 3 min radial Bunsen burner flame at 9 cm, and cubic samples at 8–10 cm of a Bunsen burner flame) and mechanical properties (via EN 1534, DIN 52185, and DIN 52186) yielded surprisingly encouraging results. Numerous samples were tested, giving the results displayed in Table 9 and Table 10. The additives of B (boric acid) and P (phosphoric acid) sources significantly improve the fire performance of the impregnated tannin solutions. We can attribute a ≈30% compressive modulus increase and ≈10% bending resistance to pine samples and ≈15% compressive modulus and ≈20% bending resistance to beech samples. The 2 min flame results, with boric acid, show a 75% and 70% decrease in flame time, a 62% and 77% decreased ember time, for pine and beech respectively, and TTIs exceeding the 120 s mark for both boric and phosphoric acids at 180 s flame exposure times; the “long” flame exposure also confirmed boric acid’s superior performance, giving a 5-fold increase in the weight loss diagram. The experiment was subsequently repeated comparing the same resin formulations above with a traditional 28 wt % DOT (disodium octaborate tetrahydrate, Na_2_B_8_O_13_ 4H_2_O) in combination with boric acid, phosphoric acid, or all three flame retardant additives via UNE 23.725–90 (dripping test), “short” and “long” exposure tests, and LOI (per ISO 4589). The extremely high B content of the DOT additive renders it the most effective formulation tested, yet it displays poor penetration and thus renders only a protective carapace while the tannin formulation offers more dimensionally consistent flame retardation. While no tannin formulation with B, P, or both as the additive(s) is likely to surpass or match an LOI of 80, they consistently demonstrate self-extinguishing behavior. Nonetheless, they do surpass the DOT’s purely superficial fire protection, which affords long-lasting flame retardancy at a potentially lower cost, albeit the authors mention the want of water resistance, appeal, and application methods for the tannin systems—see Table 11 and Table 12 [83]. 

Mimosa bark-extracted tannins, along with a variety of other tannins (see Figure 11 (Tannin Types: profisetinidin (quebracho), prorobinetinidin (mimosa), and prodelphinidin and procyanidin (pine) [79])), have a long (scientific, peer-reviewed) literary history of traditionally produced foams employing dated techniques (phenolic-based polycondensation under either acidic or alkaline conditions and subsequent physical, chemical, or combined blowing) and less-than-benign chemicals (e.g., formaldehyde, etc.) yielding surprisingly flame resistant materials comprised of nearly 95% “green” ingredients, with TTI values circa 120 s, under 50 kW/m^2^, and EHCs of 12 kW/m^2^—considerably lower than what is necessary to sustain a chain-reaction combustion, resolving self-extinguishing behaviors [83]. Inventively building upon these techniques are so-called tannin-based meringues (after the pastry; TBM) using 30–50 wt % tannin, hexamine (0.071 mass ratio to tannin, with 1.12 g *p*-toluene sulfonic acid catalyst), and 2.3 wt % Kolliphor^®^ ELP (castor-oil based surfactant; or Tween 80, as PE-sorbitol ester) beaten (sheared mechanically) in water after [83]. Figure 12 and Figure 13, respectively demonstrate the tunability of the TBM porosity and thermo-mechanical properties—note the superbly low thermal conductivity—via sole modification (read increase) of tannin concentration. These TBM foams were also reported to be more homogenous and isotropic than vertical-growth methodologies, which gives this inexpensive process a value-added and efficiency benefit in terms of value-added products. 

Pine (*Pinus radiata*) bark has also been explored as a potential tannin-furanic foam ingredient following the logic that pine or spruce based tannins are 6–7 times more reactive than those of the above (mimosa and quebracho) woods due to their phlorogulcinol A and catechol B or pyrogallol B ring (e.g., procyanidin or prodelphinidin tannins) compositions, as observable in Figure 11 [80]. Despite the theoretical benefits inherent in their chemistries, the pine-based tannin foams (made via conventional formaldehyde aided co-blow-hardened foaming) resolved fairly weak materials at low densities and offered only marginal improvements in thermal conductivity relative to the formaldehyde-made mimosa foams—and are no comparison to the TBM foams above; compare Figure 13 and Figure 14. It can be noted that at higher densities the pine-tannin-based foams are competitive with the mimosa-tannin foams from [80], despite displaying a conversely elastic behavior, but cannot yet compete with the consistently superior properties from the [85] study. 

Pure green tea extract (GTE) from *Camellia sinensis* and catechin hydrates (GTE’s main components: epicatechin monomers—i.e., epigallocatechin-3-O-gallate and epicatechin-3-O-gallate, with the former comprising 60% of catechins) were synthesized as epoxy resins with ECH and DGEBA (digylcidyl ether of bisphenol A), respectively, and cured with isophorone diamine (IPD; giving GEGTE and GEC respectively) to compare the effectiveness of the direct use of natural tannin extract (GEGTE) versus its isolate (GEC) and traditional DGEBA-IPD epoxies in terms of reactivities for future synthesis [85]. The combined results of comparison characterizations demonstrate that either (GEC or GEGTE) bio-based resin out-performs the traditional epoxy in reactivity, (in)-solubility (i.e., resistance to water), and thermal resistance (even accounting for a lower-temperature starting point), and with similar *T*_g_ values—essentially enabling scaled-up processing [86]. Of note is that fact that the pure green tea extract was sufficient, lending credibility to the hypothesis that waste tea leaves could be an inexpensive material source option, while not interfering with the food production cycle.

Polydopamine is another phenolic-rich material, derived from three chemical steps from naturally occurring dopamine. This polyphenol has the advantage of bringing nitrogen groups to the combustion zone, and has been shown to be an effective FR coating material which can be used, for example, on polyurethane foams. The polydopamine is not highly flammable, which differs from tannic acid, but this coating nonetheless is a good source of thermally-stable char [87].

## 3. Conclusions

In the progression of this study, several important and novel flame retardants were evaluated per their literature reports. Structural molecular components, architectures, processing technologies, and synergistic applications have been summarized. This article aims to provide a summarized basis for the continuation of the exploration of these and new bio-sourced flame retardants, while demonstrating the need to develop their practical applications. Chief among the features that may pre-dispose a chemical structure to flame retardation are: P and N content in the backbone or as additives, Si or silica in the active structure or as a surface coating, S content in side chains or backbone, and C bonds that decompose into ordered graphitic carbons or directly to inert gases. Areas wherein these systems must further developed include: resilience to surfactants, washing treatments (especially for fabrics per ISO 6330), employment of current large-scale manufacturing equipment (up-scaling testing), correlative studies to structure-property relationships, and large-scale testing of representative commercial products using advanced BEST-FR systems. The field of bio-based flame retardants for commercial polymers is experiencing explosive growth, which should continue for the coming decade. In order to displace the current halogenated and phosphorous-based additives, a deeper understanding of the mechanisms of flame retardation by these naturally-occurring materials will be needed; this will allow for systematic use of supplemental additives, such as clays and metal salt hydrates, needed to optimize the fire safety of finished products. Many of the bio-based agents under investigation at the present time are widely available in nature, but that is not the same thing as those substances being commercially available at acceptable market prices—as optimal additives are identified, there will be need for the chemical industry to scale up commercial isolation and purification of these products. Also of importance is that virtually all bio-based materials are hydrophilic in nature, having been produced in aqueous systems; the target polymers are conversely hydrophobic, necessitating the incorporation of optimized compatibilization additives which will produce finished compounded products which possess favorable thermal and mechanical properties.

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
