# Peer review of "Biomolecules as Flame Retardant Additives for Polymers: A Review"

_polymers, 2020, doi:10.3390/polym12040849_

Round 1
Reviewer 1 Report
In this manuscript, authors reviewed the current state-of-art of the biomolecules for the usage of flame retardant additives. Authors illustrated rich examples and made discussions on them. However, some issues still need to be well addressed.
- Nanoparticles, including nanoclay, carbon nanotubes, POSS, graphene, LDH and MXenes, as flame retardants have been extensively reported, which is an important part in the flame retardant field. Please supplement the review in the introduction section.
- Phytates, derived from beans and cereal grains (such as corn, wheat and sorghum), have been used as bio-based flame retardants for polymers. Please refer to the following articles: 1) Polymer Degradation and Stability, 2015, 119: 217-227; 2) Composites Part A: Applied Science and Manufacturing, 2018, 110: 227-236; 3) Carbohydrate Polymers, 2017, 175: 636-644.
- It is better to delete the captions appeared in the figures (e.g. Figure 3-6, 8, 9, 24). All the tables should be summarized by the authors, rather than directly copying from the references. Please pay serious attention to these figures and tables.
- There are some misused words. Please carefully check the manuscript. For example, in page 2, line 57, “O O (03:30)! CHgCHgPJNHCHgOH” is not correct. It is better to use wt.% to substitute %wt.
Author Response
Thank you for your comments and suggestions - we agree with you. For the carbon nanotubes, graphenes and clay these are beyond the scope of the bio-based FR agents we discuss, but nonetheless are important components in formulated FR products. We have inserted a statement to this effect. Someone should write a current review on these FR additives in of themselves!
We have updated the references cited by a large number, choosing these we think to be most germane to our discussion. Those references include several authors' work on phytic acid and phytates, as you suggest thank you. We have included gelatin as well.
The Figures have been cropped to remove the original captions - thank you, good catch.
We have hopefully caught all of the typographical errors, include the one you mentioned.
Reviewer 2 Report
the use of biomolecules to design fire retardant systems is a subject of major interest, which explains why a certain number of reviews have been written in recent years on it (ex: Biomacromolecules and Bio-Sourced Products for the Design of Flame Retarded Fabrics: Current State of the Art and Future Perspectives, Molecules. 2019 Oct; 24(20): 3774.)
The authors of the review entitled "Biomolecules as Flame Retardant Additives for Polymers: A Review" have chosen not to present an exhaustive review on each of the biomolecules discussed. However, it is unfortunate that only a dozen references were published after 2015 and that none relate to the last three years.
Nevertheless, the article gives a certain amount of interesting information but it would have been appreciated by the reader if the authors proposed summarizing tables or figures in order to put the different studies into perspective. This would have highlighted what is announced in the abstract.
Although this article is not the easiest to read (an additional effort of synthesis would have been appreciated) and that the last years are not covered, I suggest that it be published in its current form.
Author Response
Thank you for your comments and suggestions. Yes, we needed to, and have updated the references to include many from the past three years - thank you - that was indeed necessary.
We have provided a scheme to help the reader follow the organization of the manuscript, have cropped the figures to remove original captions, and have corrected a number of typographical errors contained in the original draft.
Round 2
Reviewer 1 Report
The manuscript has been improved by the authors. However, it is recommended that all the tables should be summarized by the authors, rather than directly copying from the articles. In addition, polydopamine (PDA) as a bio-based flame retardant has been confirmed to be effective for polymers. Please supplement the review on PDA.
Author Response
Thank you for your suggestions. Polydopamine is a bit of a stretch as a bio-based material, but it can be produced in three steps from dopa. More importantly this material does teach another really interesting lesson to readers of the review, so we have included this in the phenolic material section, citing work from the University of Texas as exemplary of the concept. On the tables reproduced in figures, we agree that some were excessively cut/pasted. Some of these tables have been re-written to summarize the data, some were edited/cropped to get to the essence of information of interest, and a few were left as-as, where the table contained data critical to the narrative on the paper. We hope this is acceptable.
Dave Schiraldi
Round 3
Reviewer 1 Report
The manuscript has been improved by the authors. It can be accepted for publication.
Author Response
Please see the response provided to the editor